# On the relation between the sharpest directions of DNN loss and the SGD step length

Stanisław Jastrzębski[1,2], Zachary Kenton[2], Nicolas Ballas[3], Asja Fischer[4], Yoshua Bengio[2,5], and Amos Storkey[6]

[1]Jagiellonian University
[2]Mila / University of Montreal
[3]Facebook AI Research
[4]Ruhr University Bochum
[5]CIFAR Senior Fellow
[6]School of Informatics, University of Edinburgh

## Abstract

Stochastic Gradient Descent (SGD) based training of neural networks with a large learning rate or a small batch-size typically ends in well-generalizing, flat regions of the weight space, as indicated by small eigenvalues of the Hessian of the training loss. However, the curvature along the SGD trajectory is poorly understood. An empirical investigation shows that initially SGD visits increasingly sharp regions, reaching a maximum sharpness determined by both the learning rate and the batch-size of SGD. When studying the SGD dynamics in relation to the sharpest directions in this initial phase, we find that the SGD step is large compared to the curvature and commonly fails to minimize the loss along the sharpest directions. Furthermore, using a reduced learning rate along these directions can improve training speed while leading to both sharper and better generalizing solutions compared to vanilla SGD. In summary, our analysis of the dynamics of SGD in the subspace of the sharpest directions shows that they influence the regions that SGD steers to (where larger learning rate or smaller batch size result in wider regions visited), the overall training speed, and the generalization ability of the final model.

## 1 Introduction

Deep Neural Networks (DNNs) are often massively over-parameterized (Zhang et al., 2016), yet show state-of-the-art generalization performance on a wide variety of tasks when trained with Stochastic Gradient Descent (SGD). While understanding the generalization capability of DNNs remains an open challenge, it has been hypothesized that SGD acts as an implicit regularizer, limiting the complexity of the found solution (Poggio et al., 2017; Advani and Saxe, 2017; Wilson et al., 2017; Jastrzębski et al., 2017).

Various links between the curvature of the final minima reached by SGD and generalization have been studied (Murata et al., 1994; Neyshabur et al., 2017). In particular, it is a popular view that models corresponding to *wide minima* of the loss in the parameter space generalize better than those corresponding to *sharp minima* (Hochreiter and Schmidhuber, 1997; Keskar et al., 2016; Jastrzębski et al., 2017; Wang et al., 2018). The existence of this empirical correlation between the curvature of the final minima and generalization motivates our study.

Our work aims at understanding the interaction between SGD and the sharpest directions of the loss surface, i.e. those corresponding to the largest eigenvalues of the Hessian. In contrast to studies such as those by Keskar et al. (2016) and Jastrzębski et al. (2017) our analysis focuses on the whole training trajectory of SGD rather than just on the endpoint. We will show in Sec. 3.1 that the evolution of the largest eigenvalues of the Hessian follows a

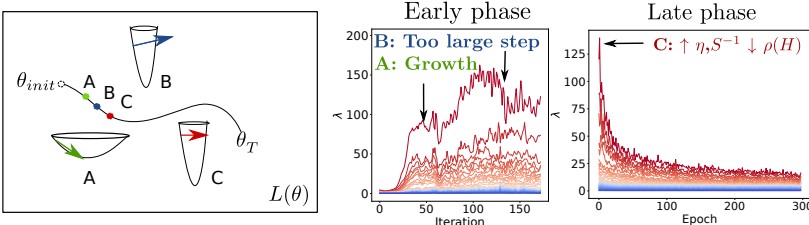

Figure 1: **Left:** Outline of the phenomena discussed in the paper. Curvature along the sharpest direction(s) initially grows (A to C). In most iterations, we find that SGD crosses the *minimum* if restricted to the subspace of the sharpest direction(s) by taking *a too large step* (B and C). Finally, curvature stabilizes or decays with a peak value determined by learning rate and batch size (C, see also right). **Right two:** Representative example of the evolution of the top 30 (decreasing, red to blue) eigenvalues of the Hessian for a SimpleCNN model during training (with $\eta = 0.005$, note that $\eta$ is close to $\frac{1}{\lambda_{max}} = \frac{1}{160}$).

consistent pattern for the different networks and datasets that we explore. Initially, SGD is in a region of broad curvature, and as the loss decreases, SGD visits regions in which the top eigenvalues of the Hessian are increasingly large, reaching a peak value with a magnitude influenced by both learning rate and batch size. After that point in training, we typically observe a decrease or stabilization of the largest eigenvalues.

To further understand this phenomenon, we study the dynamics of SGD in relation to the sharpest directions in Sec. 3.2 and Sec. 3.3. Projecting to the sharpest directions[1], we see that the regions visited in the beginning resemble bowls with curvatures such that an SGD step is typically too large, in the sense that an SGD step cannot get near the minimum of this bowl-like subspace; rather it steps from one side of the bowl to the other, see Fig. 1 for an illustration.

Finally in Sec. 4 we study further practical consequences of our observations and investigate an SGD variant which uses a reduced and fixed learning rate along the sharpest directions. In most cases we find this variant optimizes faster and leads to a sharper region, which generalizes the same or better compared to vanilla SGD with the same (small) learning rate. While we are not proposing a practical optimizer, these results may open a new avenue for constructing effective optimizers tailored to the DNNs' loss surface in the future.

On the whole this paper exposes and analyses SGD dynamics in the subspace of the sharpest directions. In particular, we argue that the SGD dynamics along the sharpest directions influence the regions that SGD steers to (where larger learning rate or smaller batch size result in wider regions visited), the training speed, and the final generalization capability.

## 2 Experimental setup and notation

We perform experiments mainly on Resnet-32[2] and a simple convolutional neural network, which we refer to as SimpleCNN (details in the Appendix D), and the CIFAR-10 dataset (Krizhevsky et al.). SimpleCNN is a 4 layer CNN, achieving roughly 86% test accuracy on the CIFAR-10 dataset. For training both of the models we use standard data augmentation on CIFAR-10 and for Resnet-32 L2 regularization with coefficient 0.005. We additionally investigate the training of VGG-11 (Simonyan and Zisserman, 2014) on the CIFAR-10 dataset (we adapted the final classification layers for 10 classes) and of a bidirectional Long Short Term Memory (LSTM) model (following the "small" architecture employed by Zaremba et al. (2014), with added dropout regularization of 0.1) on the Penn Tree Bank

---

[1]That is considering $\boldsymbol{g}_i = <\boldsymbol{g}, \boldsymbol{e}_i> \boldsymbol{e}_i$ for different $i$, where $\boldsymbol{g}$ is the gradient and $\boldsymbol{e}_i$ is the $i^{th}$ normalized eigenvector corresponding to the $i^{th}$ largest eigenvalue of the Hessian.

[2]In Resnet-32 we omit Batch-Normalization layers due to their interaction with the loss surface curvature (Bjorck et al., 2018) and use initialization scaled by the depth of the network (Taki, 2017). Additional results on Batch-Normalization are presented in the Appendix

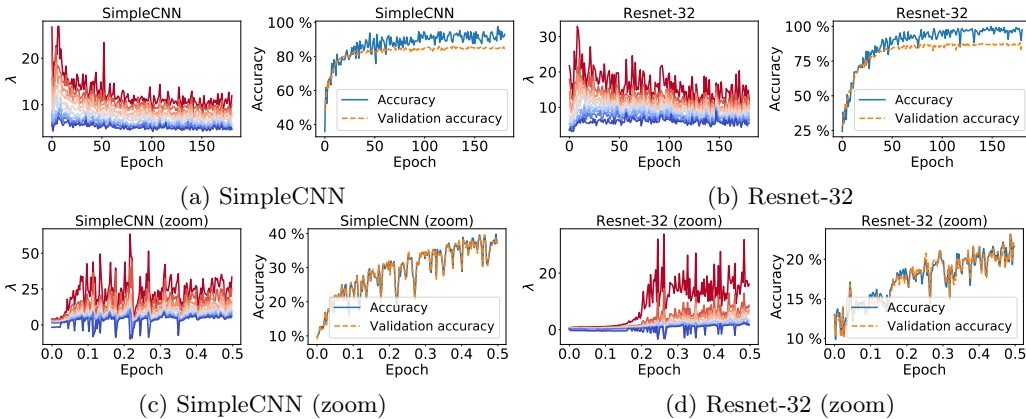

Figure 2: **Top**: Evolution of the top 10 eigenvalues of the Hessian for SimpleCNN and Resnet-32 trained on the CIFAR-10 dataset with $\eta = 0.1$ and $S = 128$. **Bottom**: Zoom on the evolution of the top 10 eigenvalues in the beginning of training. A sharp initial growth of the largest eigenvalues followed by an oscillatory-like evolution is visible. Training and test accuracy of the corresponding models are provided for reference.

(PTB) dataset. All models are trained using SGD, without using momentum, if not stated otherwise.

The notation and terminology we use in this paper are now described. We will use $t$ (time) to refer to epoch or iteration, depending on the context. By $\eta$ and $S$ we denote the SGD learning rate and batch size, respectively. $\mathbf{H}$ is the Hessian of the empirical loss at the current $D$-dimensional parameter value evaluated on the training set, and its eigenvalues are denoted as $\lambda_i$, $i = 1 \dots D$ (ordered by decreasing absolute magnitudes). $\lambda_{max} = \lambda_1$ is the maximum eigenvalue, which is equivalent to the spectral norm of $\mathbf{H}$. The top $K$ eigenvectors of $\mathbf{H}$ are denoted by $\boldsymbol{e}_i$, for $i \in \{1, \dots, K\}$, and referred to in short as the *sharpest directions*. We will refer to the mini-batch gradient calculated based on a batch of size $S$ as $\boldsymbol{g}^{(S)}(t)$ and to $\eta\boldsymbol{g}^{(S)}(t)$ as the *SGD step*. We will often consider the projection of this gradient onto one of the top eigenvectors, given by $\tilde{\boldsymbol{g}}_i(t) = \tilde{g}_i(t)\boldsymbol{e}_i(t)$, where $\tilde{g}_i(t) \equiv \langle \boldsymbol{g}^{(S)}(t), \boldsymbol{e}_i(t)\rangle$. Computing the full spectrum of the Hessian $\mathbf{H}$ for reasonably large models is computationally infeasible. Therefore, we approximate the top $K$ (up to 50) eigenvalues using the Lanczos algorithm (Lanczos, 1950; Dauphin et al., 2014), an extension of the power method, on approximately 5% of the training data (using more data was not beneficial). When regularization was applied during training (such as dropout, L2 or data augmentation), we apply the same regularization when estimating the Hessian. This ensures that the Hessian reflects the loss surface accessible to SGD. The code for the project is made available at https://github.com/kudkudak/dnn_sharpest_directions.

## 3 A study of the Hessian along the SGD path

In this section, we study the eigenvalues of the Hessian of the training loss along the SGD optimization trajectory, and the SGD dynamics in the subspace corresponding to the largest eigenvalues. We highlight that SGD steers from the beginning towards increasingly sharp regions until some maximum is reached; at this peak the SGD step length is large compared to the curvature along the sharpest directions (see Fig. 1 for an illustration). Moreover, SGD visits flatter regions for a larger learning rate or a smaller batch-size.

### 3.1 Largest eigenvalues of the Hessian along the SGD path

We first investigate the training loss curvature in the sharpest directions, along the training trajectory for both the SimpleCNN and Resnet-32 models.

**Largest eigenvalues of the Hessian grow initially.** In the first experiment we train SimpleCNN and Resnet-32 using SGD with $\eta = 0.1$ and $S = 128$ and estimate the 10 largest eigenvalues of the Hessian, throughout training. As shown in Fig. 2 (top) the spectral norm (which corresponds to the largest eigenvalue), as well as the other tracked eigenvalues, grows in the first epochs up to a maximum value. After reaching this maximum value, we observe a relatively steady decrease of the largest eigenvalues in the following epochs.

To investigate the evolution of the curvature in the first epochs more closely, we track the eigenvalues at each iteration during the beginning of training, see Fig. 2 (bottom). We observe that initially the magnitudes of the largest eigenvalues grow rapidly. After this initial growth, the eigenvalues alternate between decreasing and increasing; this behaviour is also reflected in the evolution of the accuracy. This suggests SGD is initially driven to regions that are difficult to optimize due to large curvature.

To study this further we look at a full-batch gradient descent training of Resnet-32[3]. This experiment is also motivated by the instability of large-batch size training reported in the literature, as for example by Goyal et al. (2017). In the case of Resnet-32 (without Batch-Normalization) we can clearly see that the magnitude of the largest eigenvalues of the Hessian grows initially, which is followed by a sharp drop in accuracy suggesting instability of the optimization, see Fig. 3. We also observed that the instability is partially solved through use of Batch-Normalization layers, consistent with the findings of Bjorck et al. (2018), see Fig. 9 in Appendix. Finally, we report some additional results on the late phase of training, e.g. the impact of learning rate schedules, in Fig. 12 in the Appendix.

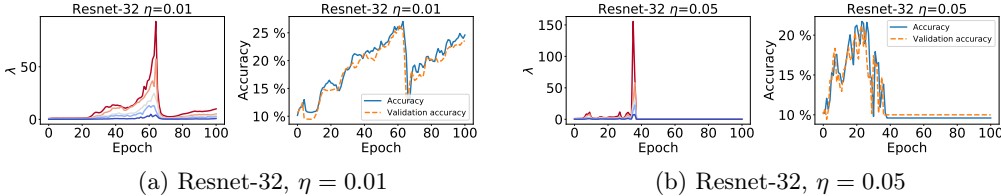

(a) Resnet-32, $\eta = 0.01$    (b) Resnet-32, $\eta = 0.05$

Figure 3: Full batch-size training of Resnet-32 for $\eta = 0.01$ and (left) $\eta = 0.05$ (right) on CIFAR-10. Evolution of the top 10 eigenvalues of the Hessian and accuracy are shown in each case. The training is unstable; an initial growth of curvature scale is followed by a sudden drop in accuracy. The CIFAR-10 dataset was subsampled to 2500 points.

**Learning rate and batch-size limit the maximum spectral norm.** Next, we investigate how the choice of learning rate and batch size impacts the SGD path in terms of its curvatures. Fig. 4 shows the evolution of the two largest eigenvalues of the Hessian during training of the SimpleCNN and Resnet-32 on CIFAR-10, and an LSTM on PTB, for different values of $\eta$ and $S$. We observe in this figure that a larger learning rate or a smaller batch-size correlates with a smaller and earlier peak of the spectral norm and the subsequent largest eigenvalue. Note that the phase in which curvature grows for low learning rates or large batch sizes can take many epochs. Additionally, momentum has an analogous effect – using a larger momentum leads to a smaller peak of spectral norm, see Fig. 13 in Appendix. Similar observations hold for VGG-11 and Resnet-32 using Batch-Normalization, see Appendix A.1.

**Summary.** These results show that the learning rate and batch size not only influence the SGD endpoint maximum curvature, but also impact the whole SGD trajectory. A high learning rate or a small batch size limits the maximum spectral norm along the path found by SGD from the beginning of training. While this behavior was observed in all settings examined (see also the Appendix), future work could focus on a theoretical analysis, helping to establish the generality of these results.

---

[3]To avoid memory limitations we preselected the first 2560 examples of CIFAR-10 to simulate full-batch gradient training for this experiment.

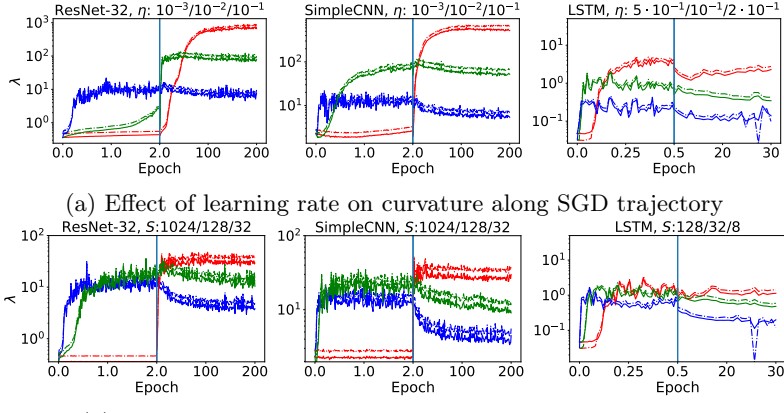

(a) Effect of learning rate on curvature along SGD trajectory

(b) Effect of batch size on curvature along SGD trajectory

Figure 4: Evolution of the two largest eigenvalues (solid and dashed line) of the Hessian for Resnet-32, SimpleCNN, and LSTM (trained on the PTB dataset) models on a log-scale for different learning rates (top) and batch-sizes (bottom). Blue/green/red correspond to increasing $\eta$ and decreasing $S$ in each figure. Left side of the vertical blue bar in each plot corresponds to the early phase of training. Larger learning rate or a smaller batch-size correlates with a smaller and earlier peak of the spectral norm and the next largest eigenvalue.

## 3.2 SHARPEST DIRECTION AND SGD STEP

The training dynamics (which later we will see affect the speed and generalization capability of learning) are significantly affected by the evolution of the largest eigenvalues discussed in Section 3.1. To demonstrate this we study the relationship between the SGD step and the loss surface shape in the sharpest directions. As we will show, SGD dynamics are largely coupled with the shape of the loss surface in the sharpest direction, in the sense that when projected onto the sharpest direction, the typical step taken by SGD is too large compared to curvature to enable it to reduce loss. We study the same SimpleCNN and Resnet-32 models as in the previous experiment in the first 6 epochs of training with SGD with $\eta$=0.01 and $S = 128$.

**The sharpest direction and the SGD step.** First, we investigate the relation between the SGD step and the sharpest direction by looking at how the loss value changes on average when moving from the current parameters taking a step only along the sharpest direction - see Fig. 6 left. For all training iterations, we compute $\mathbb{E}[L(\boldsymbol{\theta}(t) - \alpha\eta\tilde{\boldsymbol{g}}_1(t))] - L(\boldsymbol{\theta}(t))$, for $\alpha \in \{0.25, 0.5, 1, 2, 4\}$; the expectation is approximated by an average over 10 different mini-batch gradients. We find that $\mathbb{E}[L(\boldsymbol{\theta}(t) - \alpha\eta\tilde{\boldsymbol{g}}_1(t))]$ increases relative to $L(\boldsymbol{\theta}(t))$ for $\alpha > 1$, and decreases for $\alpha < 1$. More specifically, for SimpleCNN we find that $\alpha = 2$ and $\alpha = 4$ lead to a 2.1% and 11.1% increase in loss, while $\alpha = 0.25$ and $\alpha = 0.5$ both lead to a decrease of approximately 2%. For Resnet-32 we observe a 3% and 13.1% increase for $\alpha = 2$ and $\alpha = 4$, and approximately a 0.5% decrease for $\alpha = 0.25$ and $\alpha = 0.5$, respectively. The observation that SGD step does not minimize the loss along the sharpest direction suggests that optimization is ineffective along this direction. This is also consistent with the observation that learning rate and batch-size limit the maximum spectral norm of the Hessian (as both impact the SGD step length).

These dynamics are important for the overall training due to a high alignment of SGD step with the sharpest directions. We compute the average cosine between the mini-batch gradient $\mathbf{g}^S(t)$ and the top 5 sharpest directions $\mathbf{e}_1(t), \dots \mathbf{e}_5(t)$. We find the gradient to be highly aligned with the sharpest direction, that is, depending on $\eta$ and model the maximum average cosine is roughly between 0.2 and 0.4. Full results are presented in Fig. 5.

**Qualitatively, SGD step crosses the minimum along the sharpest direction.** Next, we qualitatively visualize the loss surface along the sharpest direction in the first few epochs of training, see Fig 6 (right). To better reflect the relation between the sharpest

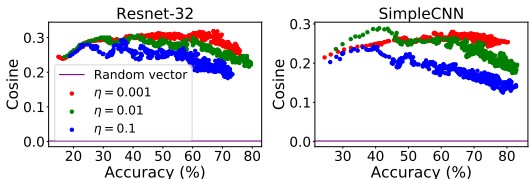

Figure 5: Average cosine between mini-batch gradient (y axis) and top sharpest directions (averaged over top 5) for different $\eta$ (color) evaluated at different level of accuracies, during training (x axis). For comparison, the horizontal purple line is alignment with a random vector in the parameter space. Curves were smoothed for clarity.

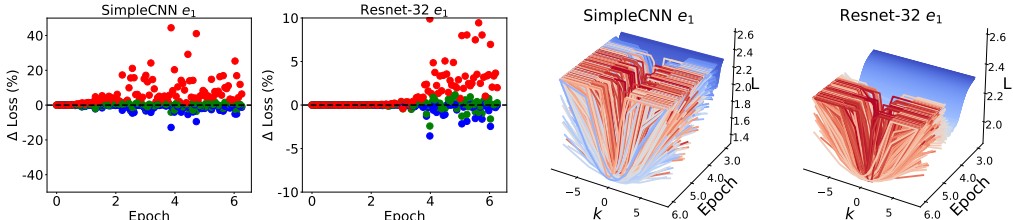

Figure 6: Early on in training, SGD finds a region such that in the sharpest direction, the SGD step length is often too large compared to curvature. Experiments on SimpleCNN and Resnet-32, trained with $\eta = 0.01$, $S = 128$, learning curves are provided in the Appendix. **Left two**: Average change in loss $\mathbb{E}[L(\boldsymbol{\theta}(t) - \alpha\eta\tilde{\boldsymbol{g}}_1(t))] - L(\boldsymbol{\theta}(t))$, for $\alpha = 0.5, 1, 2$ corresponding to red, green, and blue, respectively. On average, the SGD step length in the direction of the sharpest direction does not minimize the loss. The red points further show that increasing the step size by a factor of two leads to increasing loss (on average). **Right two**: Qualitative visualization of the surface along the top eigenvectors for SimpleCNN and Resnet-32 support that SGD step length is large compared to the curvature along the top eigenvector. At iteration $t$ we plot the loss $L(\boldsymbol{\theta}(t) + ke_1\overline{\Delta\boldsymbol{\theta}}_1(t))$, around the current parameters $\boldsymbol{\theta}(t)$, where $\overline{\Delta\boldsymbol{\theta}}_1(t)$ is the expected norm of the SGD step along the top eigenvector $e_1$. The $x$-axis represents the interpolation parameter $k$, the $y$-axis the epoch, and the $z$-axis the loss value, the color indicated spectral norm in the given epoch (increasing from blue to red).

direction and the SGD step we scaled the visualization using the expected norm of the SGD step $\overline{\Delta\boldsymbol{\theta}}_1(t) = \eta\mathbb{E}(|\tilde{g}_1(t)|)$ where the expectation is over 10 mini-batch gradients. Specifically, we evaluate $L(\boldsymbol{\theta}(t) + ke_1\overline{\Delta\boldsymbol{\theta}}_1(t))$, where $\boldsymbol{\theta}(t)$ is the current parameter vector, and $k$ is an interpolation parameter (we use $k \in [-5, 5]$). For both SimpleCNN and Resnet-32 models, we observe that the loss *on the scale of* $\overline{\Delta\boldsymbol{\theta}}_1(t)$ starts to show a bowl-like structure in the largest eigenvalue direction after six epochs. This further corroborates the previous result that SGD step length is large compared to curvature in the sharpest direction.

Training and validation accuracy are reported in Appendix B. Furthermore, in the Appendix B we demonstrate that a similar phenomena happens along the lower eigenvectors, for different $\eta$, and in the later phase of training.

**Summary.** We infer that SGD steers toward a region in which the SGD step is highly aligned with the sharpest directions and would on average increase the loss along the sharpest directions, if restricted to them. This in particular suggests that *optimization is ineffective along the sharpest direction*, which we will further study in Sec. 4.

### 3.3 How SGD steers to sharp regions in the beginning

Here we discuss the dynamics around the initial growth of the spectral norm of the Hessian. We will look at some variants of SGD which change how the sharpest directions get optimized.

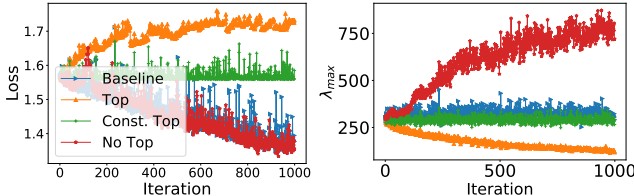

Figure 7: Evolution of the loss and the top eigenvalue during training with variations of SGD with parameter updates restricted to certain subspaces. All variants are initialized at the last iteration in Fig. 1. An SGD variant (orange) that follows at each iteration only the projection of the gradient on the top eigenvector of $\mathbf{H}$ effectively finds a region with lower $\lambda_{max}$ but increases the loss, while a variant subtracting this projection from the gradient (red), finds a sharper region while achieving a similar loss level as vanilla SGD (blue).

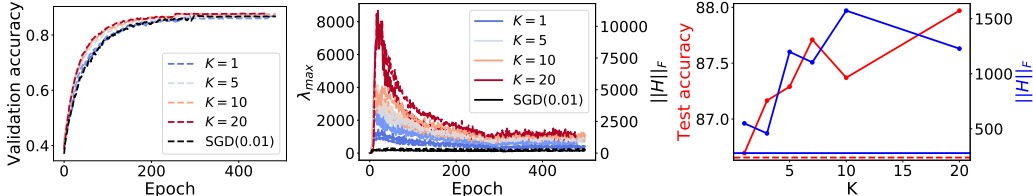

Figure 8: Nudged-SGD for an increasing number of affected sharpest directions ($K$) optimizes significantly faster, whilst generally finding increasingly sharp regions. Experiment is run using Resnet-32 and the Cifar-10 dataset. **Left and center:** Validation accuracy and the $\lambda_{max}$ and Frobenius norm (y axis, solid and dashed) using increasing $K$ (blue to red) compared against SGD baseline using the same $\eta$ (black), during training (x axis). **Rightmost:** Test accuracy (red) and Frobenius norm (blue) achieved using NSGD with an increasing $K$ (x axis) compared to an SGD baseline using the same $\eta$ (blue and red horizontal lines).

**Experiment.** We used the same SimpleCNN model initialized with the parameters reached by SGD in the previous experiment at the end of epoch 4. The parameter updates used by the three SGD variants, which are compared to vanilla SGD (blue), are based on the mini-batch gradient $\boldsymbol{g}^{(S)}$ as follows: **variant 1** (SGD top, orange) only updates the parameters based on the projection of the gradient on the top eigenvector direction, i.e. $\boldsymbol{g}(t) = \langle \boldsymbol{g}^{(S)}(t), \mathbf{e}_1(t) \rangle \mathbf{e}_1(t)$; **variant 2** (SGD constant top, green) performs updates along the constant direction of the top eigenvector $\mathbf{e}_1(0)$ of the Hessian in the first iteration, i.e. $\boldsymbol{g}(t) = \langle \boldsymbol{g}^{(S)}(t), \mathbf{e}_1(0) \rangle \mathbf{e}_1(0)$; **variant 3** (SGD no top, red) removes the gradient information in the direction of the top eigenvector, i.e. $\boldsymbol{g}(t) = \boldsymbol{g}^{(S)}(t) - \langle \boldsymbol{g}^{(S)}(t), \mathbf{e}_1(t) \rangle \mathbf{e}_1(t)$. We show results in the left two plots of Fig. 7. We observe that if we only follow the top eigenvector, we get to wider regions but don't reach lower loss values, and conversely, if we ignore the top eigenvector we reach lower loss values but sharper regions.

**Summary.** The take-home message is that SGD updates in the top eigenvector direction strongly influence the overall path taken in the early phase of training. Based on these results, we will study a related variant of SGD throughout the training in the next section.

## 4 Optimizing faster while finding a good sharp region

In this final section we study how the convergence speed of SGD and the generalization of the resulting model depend on the dynamics along the sharpest directions.

Our starting point are the results presented in Sec. 3 which show that while the SGD step can be highly aligned with the sharpest directions, on average it fails to minimize the loss if restricted to these directions. This suggests that reducing the alignment of the SGD update direction with the sharpest directions might be beneficial, which we investigate here via a variant of SGD, which we call Nudged-SGD (NSGD). Our aim here is not to build a practical

Table 1: Nudged-SGD optimizes faster and finds a sharper final endpoint with a slightly better generalization performance. Experiments on Resnet-32 model on the Cifar-10 dataset. Columns: the Frobenius norm of the Hessian at the best validation point and the final epoch; test accuracy; validation at epoch 50; cross-entropy loss in the final epoch; distance of the parameters from the initialization to the best validation epoch parameters. Experiments were performed with $\eta = 0.01$ and $S = 128$. In the last rows we report SGD using $\eta \in \{0.005, 0.1\}$.

|  | $||\mathbf{H}||_F$ | Test acc. | Val. acc. (50) | Loss | Dist. |
|---|---|---|---|---|---|
| $\gamma = 0.01$ | $1,018/1,019$ | **87.3**% | **69.3**% | 0.01324 | 21.08 |
| $\gamma = 0.1$ | $931/929$ | 87.2% | 69.0% | 0.00940 | 21.32 |
| $\gamma = 1.0$ | $272/324$ | 86.6% | 63.0% | 0.01719 | 22.90 |
| $\gamma = 5.0$ | $111/83$ | 85.5% | 52.3% | 1.17170 | 25.32 |
| SGD(0.005) | $678/946$ | 86.0% | 49.9% | 0.06221 | 18.87 |
| SGD(0.1) | $24/25$ | 88.0% | 84.7% | 0.00100 | 50.73 |

optimizer, but instead to see if our insights from the previous section can be utilized in an optimizer. NSGD is implemented as follows: instead of using the standard SGD update, $\Delta\boldsymbol{\theta}(t) = -\eta\boldsymbol{g}^{(S)}(t)$, NSGD uses a different learning rate, $\eta' = \gamma\eta$, along just the top $K$ eigenvectors, while following the normal SGD gradient along all the others directions[4]. In particular we will study NSGD with a low base learning rate $\eta$, which will allow us to capture any implicit regularization effects NSGD might have. We ran experiments with Resnet-32 and SimpleCNN on CIFAR-10. Note, that these are not state-of-the-art models, which we leave for future investigation.

We investigated NSGD with a different number of sharpest eigenvectors $K$, in the range between 1 and 20; and with the rescaling factor $\gamma \in \{0.01, 0.1, 1, 5\}$. The top eigenvectors are recomputed at the beginning of each epoch[5]. We compare the sharpness of the reached endpoint by both computing the Frobenius norm (approximated by the top 50 eigenvectors), and the spectral norm of the Hessian. The learning rate is decayed by a factor of 10 when validation loss has not improved for 100 epochs. Experiments are averaged over two random seeds. When talking about the generalization we will refer to the test accuracy at the best validation accuracy epoch. Results for Resnet-32 are summarized in Fig. 8 and Tab. 1; for full results on SimpleCNN we relegate the reader to Appendix, Tab. 2. In the following we will highlight the two main conclusions we can draw from the experimental data.

**NSGD optimizes faster, whilst traversing a sharper region.** First, we observe that in the early phase of training NSGD optimizes significantly faster than the baseline, whilst traversing a region which is an order of magnitude sharper. We start by looking at the impact of $K$ which controls the amount of eigenvectors with adapted learning rate; we test $K$ in $\{1, \ldots, 20\}$ with a fixed $\gamma = 0.01$. On the whole, increasing $K$ correlates with a significantly improved training speed and visiting much sharper regions (see Fig. 8). We highlight that NSGD with $K = 20$ reaches a maximum $\lambda_{max}$ of approximately $8 \cdot 10^3$ compared to baseline SGD reaching approximately 150. Further, NSGD retains an advantage of over 5% (1% for SimpleCNN) validation accuracy, even after 50 epochs of training (see Tab. 1).

**NSGD can improve the final generalization performance, while finding a sharper final region.** Next, we turn our attention to the results on the final generalization and sharpness. We observe from Tab. 1 that using $\gamma < 1$ can result in finding a significantly sharper endpoint exhibiting a slightly improved generalization performance compared to baseline SGD using the same $\eta = 0.01$. On the contrary, using $\gamma > 1$ led to a wider endpoint

---

[4]While NSGD can be seen as a second order method, NSGD in contrast to typical second order methods does not adapt the learning rate to be in some sense optimal given the curvature; to make it more precise we included in Appendix E a discussion on differences between NSGD and the Newton method.

[5]In these experiments each epoch of NSGD takes approximately 2-3$x$ longer compared to vanilla SGD. This overhead depends on the number of iterations needed to reach convergence in Lanczos algorithm used for computing the top eigenvectors.

and a worse generalization, perhaps due to the added instability. Finally, using a larger $K$ generally correlates with an improved generalization performance (see Fig. 8, right).

More specifically, baseline SGD using the same learning rate reached 86.4% test accuracy with the Frobenius norm of the Hessian $||\mathbf{H}||_F = 272$ (86.6% with $||\mathbf{H}||_F = 191$ on SimpleCNN). In comparison, NSGD using $\gamma = 0.01$ found endpoint corresponding to 87.0% test accuracy and $||\mathbf{H}||_F = 1018$ (87.4% and $||\mathbf{H}||_F = 287$ on SimpleCNN). Finally, note that in the case of Resnet-32 $K = 20$ leads to 88% test accuracy and $||\mathbf{H}||_F = 1100$ which closes the generalization gap to SGD using $\eta = 0.1$. We note that runs corresponding to $\eta = 0.01$ generally converge at final cross-entropy loss around 0.01 and over 99% training accuracy.

As discussed in Sagun et al. (2017) the structure of the Hessian can be highly dataset dependent, thus the demonstrated behavior of NSGD could be dataset dependent as well. In particular NSGD impact on the final generalization can be dataset dependent. In the Appendix C and Appendix F we include results on the CIFAR-100, Fashion MNIST (Xiao et al., 2017) and IMDB (Maas et al., 2011) datasets, but studies on more diverse datasets are left for future work. In these cases we observed a similar behavior regarding faster optimization and steering towards sharper region, while generalization of the final region was not always improved. Finally, we relegate to the Appendix C additional studies using a high base learning and momentum.

**Summary.** We have investigated what happens if SGD uses a reduced learning rate along the sharpest directions. We show that this variant of SGD, i.e. NSGD, steers towards sharper regions in the beginning. Furthermore, NSGD is capable of optimizing faster and finding *good generalizing sharp minima*, i.e. regions of the loss surface at the convergence which are sharper compared to those found by vanilla SGD using the same low learning rate, while exhibiting better generalization performance. Note that in contrast to Dinh et al. (2017) the sharp regions that we investigate here are the endpoints of an optimization procedure, rather than a result of a mathematical reparametrization.

## 5 Related work

**Tracking the Hessian**: The largest eigenvalues of the Hessian of the loss of DNNs were investigated previously but mostly in the late phase of training. Some notable exceptions are: LeCun et al. (1998) who first track the Hessian spectral norm, and the initial growth is reported (though not commented on). Sagun et al. (2016) report that the spectral norm of the Hessian reduces towards the end of training. Keskar et al. (2016) observe that a sharpness metric grows initially for large batch-size, but only decays for small batch-size. Our observations concern the eigenvalues and eigenvectors of the Hessian, which follow the consistent pattern, as discussed in Sec. 3.1. Finally, Yao et al. (2018) study the relation between the Hessian and adversarial robustness at the endpoint of training.

**Wider minima generalize better**: Hochreiter and Schmidhuber (1997) argued that wide minima should generalize well. Keskar et al. (2016) provided empirical evidence that the width of the endpoint minima found by SGD relates to generalization and the used batch-size. Jastrzębski et al. (2017) extended this by finding a correlation of the width and the learning rate to batch-size ratio. Dinh et al. (2017) demonstrated the existence of reparametrizations of networks which keep the loss value and generalization performance constant while increasing sharpness of the associated minimum, implying it is not just the width of a minimum which determines the generalization. Recent work further explored importance of curvature for generalization (Wen et al., 2018; Wang et al., 2018).

**Stochastic gradient descent dynamics**. Our work is related to studies on SGD dynamics such as Goodfellow et al. (2014); Chaudhari and Soatto (2017); Xing et al. (2018); Zhu et al. (2018). In particular, recently Zhu et al. (2018) investigated the importance of noise along the top eigenvector for escaping sharp minima by comparing *at the final minima* SGD with other optimizer variants. In contrast we show that from the beginning of training SGD visits regions in which SGD step is too large compared to curvature. Concurrent with this work Xing et al. (2018) by interpolating the loss between parameter values at consecutive

iterations show it is roughly-convex, whereas we show a related phenomena by investigating the loss in the subspace of the sharpest directions of the Hessian.

## 6    CONCLUSIONS

The somewhat puzzling empirical correlation between the endpoint curvature and its generalization properties reached in the training of DNNs motivated our study. Our main contribution is exposing the relation between SGD dynamics and the sharpest directions, and investigating its importance for training. SGD steers from the beginning towards increasingly sharp regions of the loss surface, up to a level dependent on the learning rate and the batch-size. Furthermore, the SGD step is large compared to the curvature along the sharpest directions, and highly aligned with them.

Our experiments suggest that understanding the behavior of optimization along the sharpest directions is a promising avenue for studying generalization properties of neural networks. Additionally, results such as those showing the impact of the SGD step length on the regions visited (as characterized by their curvature) may help design novel optimizers tailor-fit to neural networks.

## ACKNOWLEDGEMENTS

SJ was supported by grant No. DI 2014/016644 from Ministry of Science and Higher Education, Poland. SJ also received funding from projects No. 2017/25/B/ST6/01271 and No. 2017/24/T/ST6/00487 from National Science Center, Poland. Work at Mila was funded by NSERC, CIFAR, and Canada Research Chairs. This project has received funding from the European Union's Horizon 2020 research and innovation programme under grant agreement No 732204 (Bonseyes). This work is supported by the Swiss State Secretariat for Education, Research and Innovation (SERI) under contract number 16.0159.

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

## A  ADDITIONAL RESULTS FOR SEC. 3.1

### A.1  ADDITIONAL RESULTS ON THE EVOLUTION OF THE LARGEST EIGENVALUES OF THE HESSIAN

First, we show that the instability in the early phase of full-batch training is partially solved through use of Batch-Normalization layers, consistent with the results reported by Bjorck et al. (2018); results are shown in Fig. 9.

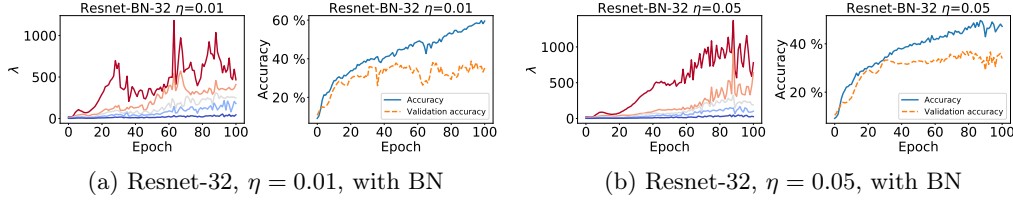

(a) Resnet-32, $\eta = 0.01$, with BN          (b) Resnet-32, $\eta = 0.05$, with BN

Figure 9: Full-batch training with Batch-Normalization is more stable. Evolution of the top 10 eigenvalues of the Hessian, and accuracy, for Resnet-32 trained on the CIFAR-10 dataset with $\eta = 0.01$ (left) and $\eta = 0.05$ (right).

Next, we extend results of Sec. 3.1 to VGG-11 and Batch-Normalized Resnet-32 models, see Fig. 11 and Fig. 10. Importantly, we evaluated the Hessian in the inference mode, which resulted in large absolute magnitudes of the eigenvalues on Resnet-32.

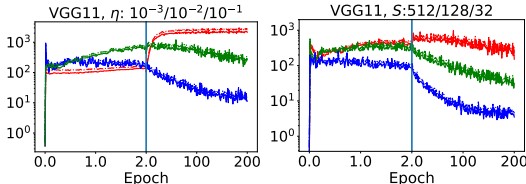

Figure 10: Same as Fig. 4, but for the VGG-11 architecture.

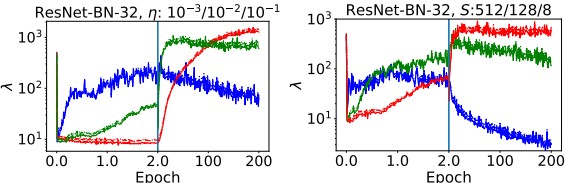

Figure 11: Same as Fig. 4, but for Resnet-32 architecture using Batch-Normalization layers.

### A.2  IMPACT OF LEARNING RATE SCHEDULE

In the paper we focused mostly on SGD using a constant learning rate and batch-size. We report here the evolution of the spectral and Frobenius norm of the Hessian when using a simple learning rate schedule in which we vary the *length* of the first stage $L$; we use $\eta = 0.1$ for $L$ epochs and drop it afterwards to $\eta = 0.01$. We test $L$ in $\{10, 20, 40, 80\}$. Results are reported in Fig. 12. The main conclusion is that depending on the learning rate schedule in the next stages of training curvature along the sharpest directions (measured either by the spectral norm, or by the Frobenius norm) can either decay or grow. Training for a shorter time (a lower $L$) led to a growth of curvature (in term of Frobenius norm and spectral norm) after the learning drop, and a lower final validation accuracy.

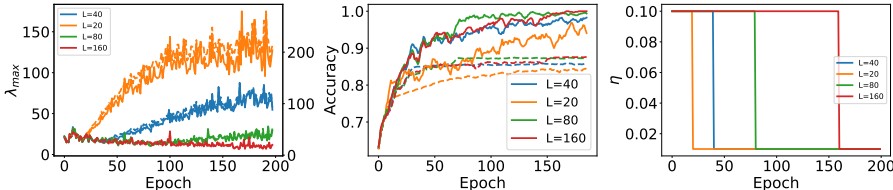

Figure 12: Depending on the length of the first stage of a learning rate schedule, spectral norm of the Hessian and generalization performance evolve differently. **Left:** Spectral norm of the Hessian (solid) and Frobenius norm (dashed), during training (x axis). **Center:** Accuracy and validation accuracy, during training (x axis). **Right:** Learning rate, during training (x axis).

### A.3 IMPACT OF USING MOMENTUM

In the paper we focused mostly on experiments using plain SGD, without momentum. In this section we report that large momentum similarly to large $\eta$ leads to a reduction of spectral norm of the Hessian, see Fig. 13 for results on the VGG11 network on CIFAR10 and CIFAR100.

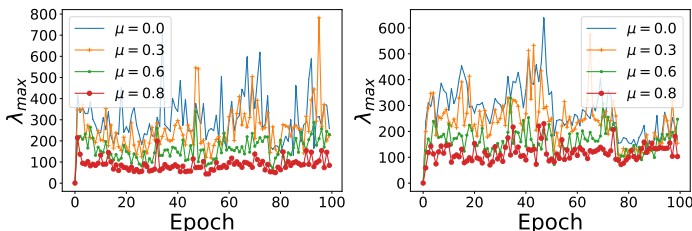

Figure 13: Momentum limits the maximum spectral norm of the Hessian throughout training. Training at $\eta = 0.1$ and $S = 128$ on VGG11 and CIFAR10 (left) and CIFAR100 (right).

## B ADDITIONAL RESULTS FOR SEC. 3.2

In Fig. 14 we report the corresponding training and validation curves for the experiments depicted in Fig. 6. Next, we plot an analogous visualization as in Fig. 6, but for the 3rd and 5th eigenvector, see Fig. 15 and Fig. 16, respectively. To ensure that the results do not depend on the learning rate, we rerun the Resnet-32 and SimpleCNN experiments with $\eta = 0.05$, see Fig. 17.

Finally, we run the same experiment as in Sec. 3.2, but instead of focusing on the early phase we replot Fig. 6 for the first 200 epochs, see Fig. 18

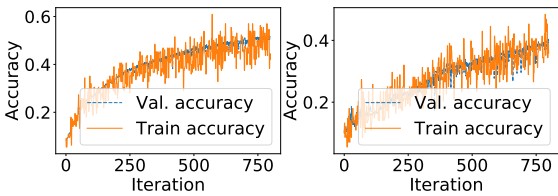

Figure 14: Training and validation accuracy for experiments in Fig 6. Left corresponds to SimpleCNN. Right corresponds to Resnet-32.

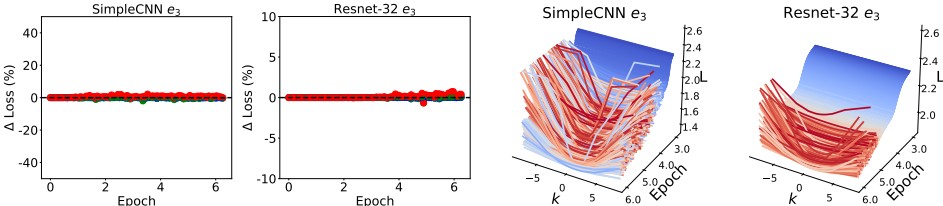

Figure 15: Similar to Fig 6 but computed for the third eigenvector of the Hessian. We see that the loss surface is much flatter in this direction. To facilitate a direct comparison, scale of each axis is kept the same as in Fig. 6.

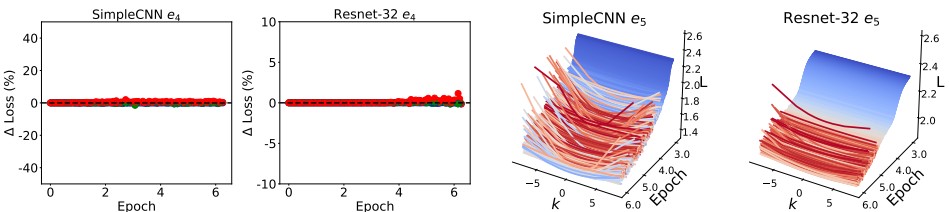

Figure 16: Similar to Fig 6 but for the fourth eigenvector. We see that the loss surface is much flatter in this direction. To facilitate a direct comparison, scale of each axis is kept the same as in Fig. 6.

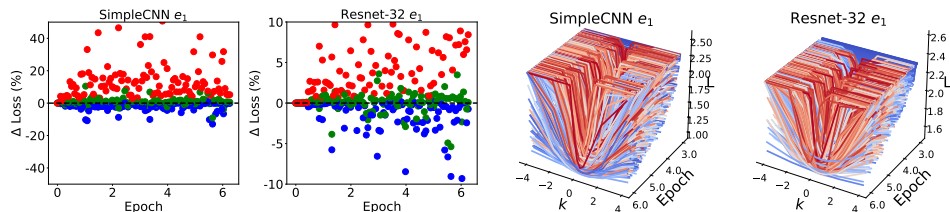

Figure 17: Similar to Fig 6 but for Resnet-32 (top) and SimpleCNN (bottom) with $\eta = 0.05$.

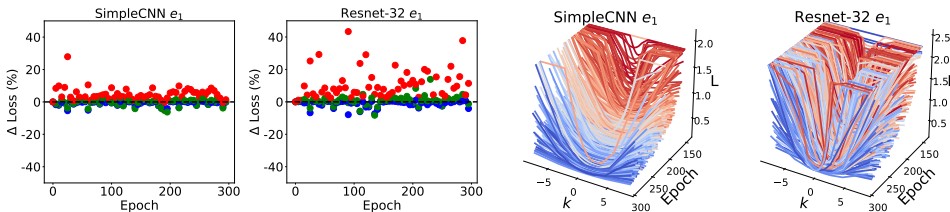

Figure 18: Similar to Fig 6 but for Resnet-32 (top) and SimpleCNN (bottom) run for 200 epochs.

## C    ADDITIONAL RESULTS FOR SEC. 4

Here we report additional results for Sec. 4. Most importantly, in Tab. 2 we report full results for SimpleCNN model. Next, we rerun the same experiment, using the Resnet-32 model, on the CIFAR-100 dataset, see Tab. 5, and on the Fashion-MNIST dataset, see Tab. 6. In the case of CIFAR-100 we observe that conclusion carry-over fully. In the case of Fashion-MNIST we observe that the final generalization for the case of $\gamma < 1$ and $\gamma = 1.0$ is

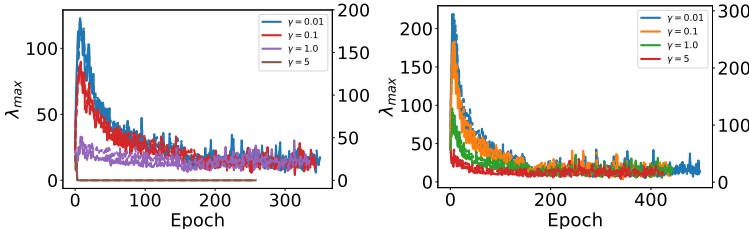

Figure 19: Evolution of the spectral norm and Frobenius norm of the Hessian (y axis, solid and dashed line, respectively) for different $\gamma$ (color) for the experiment using a larger learning (left) and momentum (right), see text for details.

similar. Therefore, as discussed in the main text, the behavior seems to be indeed dataset dependent.

In Sec. 4 we purposedly explored NSGD in the context of suboptimally picked learning rate, which allowed us to test if NSGD has any implicit regularization effect. In the next two experiments we explored how NSGD performs when using either a large base learning $\eta$, or when using momentum. Results are reported in Tab. 3 (learning rate $\eta = 0.1$) and Tab. 4 (momentum $\mu = 0.9$). In both cases we observe that NSGD improves training speed and reaches a significantly sharper region initially, see Fig. 22. However, the final region curvature in both cases, and the final generalization when using momentum, is not significantly affected. Further study of NSGD using a high learning rate or momentum is left for future work.

Table 2: Same as Tab. 1, but for Simple-CNN model. NSGD with $\gamma = 5.0$ diverged and was excluded from the Table.

|  | $\|\mathbf{H}\|_F$ | Test acc. | Val. acc. (50) | Loss | Dist. |
|---|---|---|---|---|---|
| $\gamma = 0.01$ | 423/429 | 87.1% | 83.5% | 0.00143 | 17.63 |
| $\gamma = 0.1$ | 294/268 | 87.4% | 83.3% | 0.00126 | 17.98 |
| $\gamma = 1.0$ | 127/127 | 86.6% | 82.3% | 0.00063 | 19.76 |
| $\gamma = 5.0$ | 130/129 | 85.2% | 79.8% | 0.00160 | 20.26 |
| SGD(0.005) | 476/516 | 85.8% | 78.0% | 0.00862 | 14.94 |
| SGD(0.05) | 34/26 | 87.4% | 85.1% | 0.00008 | 35.38 |

Table 3: Same as Tab. 1, but for Resnet-32 and NSGD using base learning rate $\eta = 0.1$. NSGD with $\gamma = 5.0$ diverged and was excluded from the Table.

|  | $\|\mathbf{H}\|_F$ | Test acc. | Val. acc. (50) | Loss | Dist. |
|---|---|---|---|---|---|
| $\gamma = 0.01$ | 32/34 | 89.3% | 85.8% | 0.00322 | 49.04 |
| $\gamma = 0.1$ | 20/26 | 89.2% | 85.9% | 0.00105 | 50.73 |
| $\gamma = 1.0$ | 25/19 | 87.9% | 83.2% | 0.00246 | 50.70 |

Table 4: Same as Tab. 1, but for Resnet-32 and NSGD using momentum coefficient 0.9.

|  | $\|\mathbf{H}\|_F$ | Test acc. | Val. acc. (50) | Loss | Dist. |
|---|---|---|---|---|---|
| $\gamma = 0.01$ | 20/20 | 89.4% | 86.9% | 0.00225 | 47.57 |
| $\gamma = 0.1$ | 36/32 | 89.3% | 87.1% | 0.00082 | 47.14 |
| $\gamma = 1.0$ | 27/34 | 89.2% | 85.8% | 0.00239 | 49.57 |
| $\gamma = 5.0$ | 35/44 | 88.2% | 83.9% | 0.00438 | 51.07 |

Table 5: Same as Tab. 1, but for Simple-CNN and NSGD on the Fashion-MNIST dataset.

|  | $\|\mathbf{H}\|_F$ | Test acc. | Val. acc. (50) | Loss | Dist. |
|---|---|---|---|---|---|
| $\gamma = 0.01$ | 294/431 | 92.6% | 90.3% | 0.05300 | 13.73 |
| $\gamma = 0.1$ | 257/341 | 92.4% | 90.1% | 0.04340 | 13.62 |
| $\gamma = 1.0$ | 159/162 | 92.5% | 89.8% | 0.04672 | 14.85 |
| $\gamma = 5.0$ | 75/101 | 91.7% | 89.2% | 0.07820 | 14.65 |

Table 6: Same as Tab. 1, but for Resnet-32 and NSGD on the CIFAR-100 dataset.

|  | $\|\mathbf{H}\|_F$ | Test acc. | Val. acc. (50) | Loss | Dist. |
|---|---|---|---|---|---|
| $\gamma = 0.01$ | $1,315/1,544$ | 56.9% | 27.8% | 0.18048 | 31.37 |
| $\gamma = 0.1$ | $797/1,192$ | 57.4% | 27.0% | 0.29081 | 31.18 |
| $\gamma = 1.0$ | 470/720 | 54.1% | 25.6% | 0.28299 | 32.67 |
| $\gamma = 5.0$ | 240/378 | 53.2% | 20.6% | 0.51348 | 34.20 |

## D  SimpleCNN Model

The SimpleCNN used in this paper has four convolutional layers. The first two have 32 filters, while the third and fourth have 64 filters. In all convolutional layers, the convolutional kernel window size used is (3,3) and 'same' padding is used. Each convolutional layer is followed by a ReLU activation function. Max-pooling is used after the second and fourth convolutional layer, with a pool-size of (2,2). After the convolutional layers there are two linear layers with output size 128 and 10 respectively. After the first linear layer ReLU activation is used. After the final linear layer a softmax is applied. Please also see the provided code.

## E  Comparing NSGD to Newton Method

Nudged-SGD is a second order method, in the sense that it leverages the curvature of the loss surface. In this section we argue that it is significantly different from the Newton method, a representative second order method.

The key reason for that is that, similarly to SGD, NSGD is driven to a region in which curvature *is too large compared to its typical step*. In other words *NSGD does not use an optimal learning rate for the curvature, which is the key design principle for second order methods.* This is visible in Fig. 20, where we report results of a similar study as in Sec. 3.2, but for NSGD ($K = 5$, $\gamma = 0.01$). The loss surface appears sharper in this plot, because reducing gradients along the top $K$ sharpest directions allows optimizing over significantly sharper regions.

As discussed, the key difference stems for the early bias of SGD to reach maximally sharp regions. It is therefore expected that in case of a quadratic loss surface Newton and NSGD optimizers are very similar. In the following we construct such an example. First, recall that update in Newton method is typically computed as:

$$\boldsymbol{\theta}(t + 1) = \boldsymbol{\theta}(t) - \eta(H + \lambda I)^{-1} g^S(\boldsymbol{\theta}(t)), \tag{1}$$

where $\lambda$ is a scalar. Now, if we assume that $H$ is diagonal and put $diag(H) = [100, 100, 100, 100, 100, 1, 1]$, and finally let $\lambda = 0$, it can be seen that the update of NSGD with $\gamma = 0.01$ and $K = 5$ is equivalent to that of Newton method.

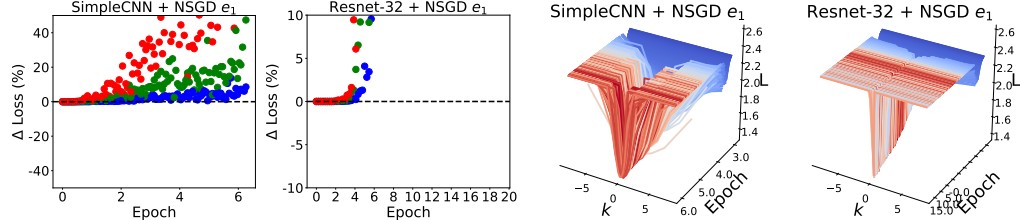

Figure 20: Same as Fig. 6, but for NSGD with $\gamma = 0.01$ and $K = 5$.

## F EXPERIMENTS ON SENTIMENT ANALYSIS DATASET

Most of the experiments in the paper focused on image classification datasets (except for language modeling experiments in Sec. 3.1). The goal of this section is to extend some of the experiments to the text domain. We experiment with the IMDB (Maas et al., 2011) binary sentiment classification dataset and use the simple CNN model from the Keras (Chollet et al., 2015) example repository[6].

First, we examine the impact of learning rate and batch-size on the Hessian along the training trajectory as in Sec. 3.1. We test $\eta \in \{0.025, 0.05, 0.1\}$ and $S \in \{2, 8, 32\}$. As in Sec. 3.1 we observe that the learning rate and the batch-size limit the maximum curvature along the training trajectory. In this experiment the phase in which the curvature grows took many epochs, in contrast to the CIFAR-10 experiments. The results are summarized in Fig. 21.

Next, we tested Nudged SGD with $\eta = 0.01$, $S = 8$ and $K = 1$. We test $\gamma \in \{0.1, 1.0, 5.0\}$. We increased the number of parameters of the base model by increasing by a factor of 2 number of filters in the first convolutional layer and the number of neurons in the dense layer to encourage overfitting. Experiments were repeated over 3 seeds.

We observe that in this setting NSGD for $\gamma < 1$ optimizes significantly faster and finds a sharper region initially. At the same time using $\gamma < 1$ does not result in finding a better generalizing region. The results are summarized in Tab. 7 and Fig. 22.

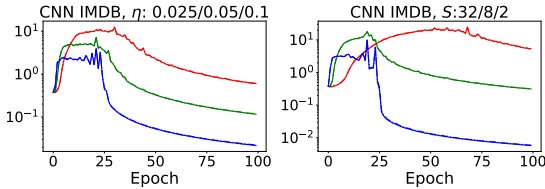

Figure 21: Same as Fig. 4, but for CNN model on the IMDB dataset.

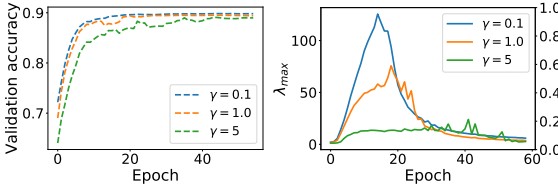

Figure 22: NSGD experiments on IMDB dataset. Evolution of the validation accuracy (left) and spectral norm of the Hessian (left) for different $\gamma$.

---

[6]Accessible at https://github.com/keras-team/keras/blob/master/examples/imdb_cnn.py

Table 7: Same as Tab. 1, but for CNN on IMDB dataset.

|  | $||\mathbf{H}||_F$ | Test acc. | Val. acc. (10) | Loss | Dist. |
|---|---|---|---|---|---|
| $\gamma = 0.1$ | 5.91/1.94 | 88.83% | 86.79 | 0.00031 | 13.24 |
| $\gamma = 1.0$ | 4.19/1.45 | 88.79% | 84.52 | 0.00029 | 14.04 |
| $\gamma = 5.0$ | 2.36/0.79 | 88.37% | 78.32 | 0.00023 | 16.85 |

