# OpenReview forum: "On the Relation Between the Sharpest Directions of DNN Loss and the SGD Step Length"
_ICLR.cc/2019/Conference_

### Official Review · AnonReviewer1 · 2018-10-22
**Great analyses about the relationship between the convergence/generalization and  the update on largest eigenvectors of Hessian of the empirical loss.**

**Rating:** 7
**Confidence:** 3

**Review:**

Updated rating after author response from 8 to 7 because I agree that Figure 1 and some discussions were confusing in the original manuscript.
--------------------------------------------------------------------------

This paper investigates the relationship between the eigenvectors of the Hessian. This paper investigates characteristics of Hessian of the empirical losses of DNNs through comprehensive experiments. These experiments showed many important insights, 1) the top-K eigenvalues become bigger in the early stage, and decrease in later stage. 2) Bigger SGD steps and smaller batch-size leads to smaller and earlier peak of eigenvalues. 3) The sharpest direction update does not contribute to the loss value decrease in the normal step size (or bigger). From these analyses, this paper proposes to decrease the SGD step length on top-K eigenvectors for speeding up the convergence. Experimental results showed that the proposed method could converge to local minima in a fewer epoch and obtain better result, which means higher test accuracy.

This paper is well-written and well-organized. Findings about eigenvalues and these relationship between the SGD step length are very impressive. Although the step length adjustment on the top-K eigenvector directions are not realistic solution for improving the current SGD-based optimization on DNNs due to heavy computational cost, I think these findings and insights are very helpful to ICLR and other ML communities.

---

> ### Author Response · Authors · 2018-11-13
> **Response to Reviewer 1**
>
> We thank the reviewer for the positive feedback. We are glad that NSGD experiments were found to be an interesting investigation. Please also find a summary of results of additional experiments we conducted in response to the other reviews here: https://goo.gl/yYM1DG.
>
> EDIT: We updated now the manuscript and added a summary of the experiments with a more careful analysis of NSGD results on IMDB.

---

### Official Review · AnonReviewer2 · 2018-11-02
**see review**

**Rating:** 6
**Confidence:** 3

**Review:**

The paper discusses connections between the properties of DNN loss surfaces and the step length SGD algorithms take, a timely topic.  On the whole, reasonably well done, with some interesting observations.

It makes several claims, most notably that there is an initial regime where SGD visits increasingly sharp regions of the loss surface, followed by a regime where the loss surface gets smoother.  Useful to know, and characterized moderately well.

A weakness is that the generality of that claim is not made clear.  Like many papers in the area, it is an observation, the realm of which is not clarified.  E.g., what properties of the neural network or data does it depend on.  Also not clarified is how this depends on initialization, etc.

The evaluation should be more systematic, as it is hard to tell how general is the claims of the paper as well as how they depend on implementation details.

The discussion of Hessian directions ignores very relevant work by Yao et al (https://arxiv.org/abs/1802.08241 and follow up).

The first figure in Fig 1 is probably misleading, and probably not worth having, the latter two are what is measured and thus more interesting.

The obvious conclusion from the poor conditioning is that methods designed to addressed poor conditioning, i.e., second order methods, should be considered.  Those should have a complementary dynamics to what is discussed.  This is what is the elephant in the room when you talk about steering towards or away from regions whose curvature matches the SGD step.

I don't know what it means to say "Where applicable, the Hessian is estimated with regularization applied"  Is this to speed up computation, why doesn't this change the loss surface, etc.  If you are not measuring Hessian information precisely, then all the claims of the paper fall apart.

Several times claims like "SGD reaches a region in which the SGD step matches ..."  Of course, the energy surface changes with training time, so it is a little unclear what is being said.

The main method Nudged-SGD sounds like a poor-mans second order method.  Why not describe it as such (in more than a footnote and appendix), rather than introducing a new acronym.  I don't know that I believe the "key design principle" in the appendix for second order methods.  Second order methods rotate and stretch to take a locally-correct step length, and this method sounds like it is doing a poor mans version of that.  There is a good question as to whether the "thresholding" into large and small that NSGD is doing causes it to do something very different, but that isn't really evaluated.

Averaging over two random seeds is not a lot.

---

> ### Author Response · Authors · 2018-11-13
> **Response to Reviewer 2**
>
> We thank the reviewer for his valuable comments. Based on yours and other reviewers’ remarks we run additional experiments using Adam, different initialization schemes  and on data from a sentence classification task. We summarized them in https://goo.gl/yYM1DG, and would be happy to add them to the paper. We will address now each point in order.
>
> * On generality *
> On the whole, our experiments were run on CIFAR-10 and PTB as described  in the main text, and CIFAR-100 and Fashion-MNIST as descibed in the Appendix. We also experimented with 4 models (Resnet-32, SimpleCNN, VGG, and LSTM). We therefore believe that our main results describing how the Hessian behaves along the optimization trajectory were  supported by a reasonable (compared to similar papers in the domain) set of settings. Please also note that related results were observed in concurrent ICLR submissions [1], [2] and [3]. In particular [2] shows that indeed a measure of curvature (Fisher Information) closely related to the Hessian grows initially very quickly - which confirms some of our observations in 3.1.
>
> Having said that we fully agree that extending the analysis to different initialization and dataset dependence would be desirable. We rerun similar analysis to 3.1 using Adam, different initialization (we compared uniform to normal, with different scaling) and on IMDB (a sentence classification task). These experiment corroborate our main finings.
>
> * Extending results to second order methods *
> We fully agree that investigating second order methods would be very interesting. Based on your remark as the first step towards this direction we rerun some of the experiments using Adam, see https://goo.gl/yYM1DG. On the whole the main focus of the paper is on SGD, and thus a more extensive study perhaps should left for future work.
>
> Hessian and regularization. We apologize for the unclear formulation. We wanted to say, that we used regularization when computing the Hessian (e.g. including L2 terms, or sampling dropout mask) if this was also done for computing the loss  uring optimization. In this sense we get a  more *realistic* estimate and this choice has *no bearing on the computation speed*. We will make this more clear in the revised version of the manuscript.
>
> What does “SGD matches curvature” mean. Let us clarify what we mean by the phrase that SGD finds a region where its steps matches the curvature. Consider projecting SGD step onto the directions corresponding to the largest eigenvalues of the Hessian. Our claim is that along these directions the projection is too large to reduce the loss. Visually, SGD step crosses the minima in the subspace spanned by the sharpest directions. Please also see Fig.1 for an illustration. We agree that wording is confusing, and we will formulate this in the revised version.
>
> *NSGD as a poor-mans second order *
> We agree that NSGD is a second order method in the sense that it uses second order information to adapt the step-size. It is different from typical second order methods in that it does not seek to minimize loss along the sharpest directions. Instead, NSGD step typically crosses over the minima along the sharpest direction, just like in the case of SGD (in the sense as depicted in Fig. 1, and as discussed in the last Appendix). To further clarify - the goal of this section was to investigate importance of SGD dynamics along the sharpest directions. We did not seek to prove NSGD is a better optimizer than other second order methods, which is why we were inadvertently brief in the discussion about how it differs from other second order methods.  We will clarify all of this and in particular note that NSGD is a specific form of a second order method.
>
> * Other points *
> Thank you for pointing us to Yao et al. We will add a discussion of Yao et al. to ‘Related work’.
>
> You mentioned that Fig. 1 is not useful. In general, we would like to keep an intuitive depiction of the main findings. Please let us know if you have any suggestions how to improve Fig. 1.
> ---
>
> Thank you again for your valuable comments, and we will update the manuscript shortly.
>
> [1] Gradient Descent Happens in a Tiny Subspace, https://openreview.net/forum?id=ByeTHsAqtX
> [2] Critical Learning Periods, https://openreview.net/forum?id=BkeStsCcKQ&noteId=BkeStsCcKQ
> [3] A Walk with SGD: How SGD Explores Regions of Deep Network Loss?, https://openreview.net/forum?id=B1l6e3RcF7&noteId=BylzRFgP2Q
>
> EDIT: We updated now the manuscript and added a summary of the experiments with a more careful analysis of NSGD results on IMDB.

---

### Official Review · AnonReviewer3 · 2018-11-03
**Good idea. Not convinced about generalizability of results.**

**Rating:** 6
**Confidence:** 4

**Review:**

Update after author response: I am changing my rating from 4 to 6 in light of the clarification and new experiments.

-------
In this paper the authors study the relationship between the SGD step size and the curvature of the loss surface, empirically showing that: 1) SGD is guided towards sharp regions of the loss surface at the start especially with a large learning rate or a small batch size. 2) Loss increases on average when taking a SGD step in the sharpest directions. 3) Modifying the SGD step size in the sharp directions (for example removing its component in the sharpest direction), can lead to substantial changes in both the quality and the local landscape of the minima (for the example mentioned, leading to a better and sharper minima). Motivated by these observations, the authors propose a variant of SGD that leads to better performance on the datasets considered.

Deep learning theory is a very important frontier for machine learning and one that’s needed to make the practice be guided more by the foundational principles than incessant tweaks. The paper makes some very interesting observations and uses those insights to improve the widely used SGD. However, I have a few concerns which leave me unconvinced about the impact of the contributions in the paper. My biggest problem is the use of second order information in the algorithm which makes the optimization process computationally cumbersome, and raises the question as to why might this approach be preferable to any other second order approach (the authors touch on Newton method in the appendix but the discussion far from settles the matter). Similar questions arise in considering the merit of the proposed methods in comparison to a host of other well-studied augmentations to SGD like momentum, Adam or AdaGrad. The quality of presentation is also a problem, and both the organization of the main matter as well as of the figures can use some polishing. The latter specifically sometimes lacked legends (Fig. 3 and 4), and some other times had legends covering a quarter of the plot (Fig. 5). Lastly, even though the claims sound theoretical, they are not derived from any set of first principles but come from observations on a few datasets. While this may after all be how SGD behaves in general, currently the paper doesn’t provide any evidence to believe that.

Minor issues: “withe” (page 2, spelling), “\alpha = 0.5, 1, 2 corresponding to red, green, and blue” (page 4, I believe it should be “blue, green and red”).

In summary, even though I liked what the paper set out to do, I am not convinced on the generalizability of these results and subsequently the rationale for using the proposed method over other competing options. A revised version of the paper with either validation on more datasets or sound theory generalizing the results to some extent would make for a much nicer contribution.

---

> ### Author Response · Authors · 2018-11-13
> **Response to Reviewer 3**
>
> We thank the reviewer for the valuable comments. The biggest concerns raised are the generalizability of the experimental results and the practical applicability of the analysed SGD variant, NSGD, due to the use of second order information (the top eigenvectors of the Hessian).
>
> * Proposing a practical optimizer is not the main goal of the paper *
> First we would like to stress that proposing a practical optimizer was not the goal of the paper. Instead, our goal was to study the Hessian of the training loss along the optimization trajectory, and the relation of the SGD step to the sharpest directions. Experiments on NSGD were run to investigate the importance of this relation for optimization and generalization of neural networks. We agree that some of the formulations (like the opening sentence of Sec.4, or part of the abstract) were confusing in this respect, and we will make it more clear in the revised version.
>
> Based on the remarks we run additional experiments using Adam, different initialization schemes,  and data from  a sentence classification task (including experiments using NSGD). We summarized them in https://goo.gl/yYM1DG, and would be happy to add them to the paper based on the reviewers feedback.
>
> *Generality of results*
> Another key concern raised is about generality of the results. On the whole, our experiments were run on CIFAR-10 and PTB (results shown in the main text), CIFAR-100 and Fashion-MNIST (results shown in the Appendix). We also experimented with 4 models in total (Resnet-32, SimpleCNN, VGG, and LSTM). We however agree that extending the experiments to different datasets, network architectures and training settings is desirable. Based on the remarks we rerun some of the experiments using different initializations, and for a new sentence classification task.
>
> NSGD experiments were conducted on Fashion-MNIST, Cifar-10, Cifar-100 using SimpleCNN and ResNet32 models. The main purpose of these experiments was to show that behavior along sharpest directions can be important for training speed and generalization. We acknowledged in the text that NSGD results might be dataset dependent because the structure of the Hessian is dataset dependent (as shown for instance by Sagun et al, https://arxiv.org/abs/1706.04454). We will make it clearer in the revised version of the manuscript. We also rerun NSGD experiments on a text classification dataset.
>
> Furthermore, related results were observed in concurrent ICLR submissions [1], [2], and [3], which further supports generalizability of the results.  [1] shows that indeed gradient step is highly aligned with the Hessian from the beginning (which is one of the observations discussed in 3.2). [2] shows that indeed a measure of curvature (Fisher Information Metric) closely related to the Hessian grows initially very quickly. Finally, [3] shows a related phenomena that SGD starts to oscillate early on in training, especially for a large batch-size. [2] and [3] are consistent with our results in 3.1.
>
> * NSGD practicality *
> Finally, we agree that NSGD might be an impractical optimizer, because of its use of second order information. Note however, NSGDs overhead incurred by computing the top eigenvectors of the Hessian is comparable to that of methods like K-FAC, which are considered practical. We will clarify the writing. We also run experiments like in Sec. 3.1 with Adam  as an optimizer as a first step towards understanding how the analysis extends to methods adapting to the curvature.
>
> --
>
> All the aforementioned additional results are summarized in https://goo.gl/yYM1DG. Do you have any other experiments in mind that you would like us to run?
>
> Thank you again for your comments, and we will update the manuscript shortly.
>
> [1] Gradient Descent Happens in a Tiny Subspace, https://openreview.net/forum?id=ByeTHsAqtX
> [2] Critical Learning Periods, https://openreview.net/forum?id=BkeStsCcKQ&noteId=BkeStsCcKQ
> [3] A Walk with SGD: How SGD Explores Regions of Deep Network Loss?, https://openreview.net/forum?id=B1l6e3RcF7&noteId=BylzRFgP2Q
>
> EDIT: We updated now the manuscript and added a summary of the experiments with a more careful analysis of NSGD results on IMDB.

---

### Author Response · Authors · 2018-11-23
**Revised version: clarifications and additional experiments**

We would like to thank again the reviewers for their comments and suggestions for experiments.

Summary of the main changes to the manuscript:
  * We rephrased parts of the abstract to clarify the motivation and main findings
  * We clarified parts of the paper based on comments by R1 and R2. Most importantly, we clarified the goal and generality of the NSGD experiments. We also unified the way we refer to the relation between the SGD step and sharpest direction, which
previously was found confusing by R2.
  * Based on the suggestions by R1 and R2 we run additional experiments using Adam, different initializations and extending results of Sections 3.1 and 4 (NSGD) to sentiment classification task (https://goo.gl/yYM1DG) We included the sentiment classification results in the Appendix, and are open to including other results as well. The results are generally in line with the main text, hopefully highlighting the generality of the main claims.

Thank you,
The authors

---

### Meta-Review · Area_Chair1 · 2018-12-13
**Good but more study needed**

**Confidence:** 5
**Recommendation:** Accept (Poster)

**Metareview:**

The reviewers found the paper insightful and the authors explanations well-provided. However the paper would benefit from more systematic empirical evaluation and corresponding theoretical intuition.